# The Knockout of the *ASIP* Gene Altered the Lipid Composition in Bovine Mammary Epithelial Cells via the Expression of Genes in the Lipid Metabolism Pathway

**DOI:** 10.3390/ani12111389

**Published:** 2022-05-28

**Authors:** Tao Xie, Yinuo Liu, Huixian Lu, Ambreen Iqbal, Mengru Ruan, Ping Jiang, Haibin Yu, Jilun Meng, Zhihui Zhao

**Affiliations:** 1Department of Animal Sciences, College of Coastal Agricultural Sciences, Guangdong Ocean University, Zhanjiang 524088, China; gdou1502753565@163.com (T.X.); 15766381766@163.com (H.L.); ambreeniqbal1071@gmail.com (A.I.); 18438615080@163.com (M.R.); jiangp@gdou.edu.cn (P.J.); yuhb@gdou.edu.cn (H.Y.); 2Zhejiang Institute of Freshwater Fisheries, Huzhou 313000, China; liuyn88@hotmail.com; 3The Key Laboratory of Animal Resources and Breed Innovation in West Guangdong, Zhanjiang 524088, China

**Keywords:** bMECs, *ASIP*, CRISPR/Cas9, lipid metabolism, fatty acid

## Abstract

**Simple Summary:**

According to the FDA, a litre of milk contains about 33 g of total lipids, 95% of which are triacylglycerols made up of varying lengths and saturation of fatty acids. The increase in free fatty acids in milk may lead to changes in the flavour of milk and milk products. Bovine mammary epithelial cells (bMECs) function to synthesize and secrete milk and are good in vitro models for cells in milk quality research. In this research, we report for the first time the effect of the ASIP gene knockout in bovine mammary epithelial cells using CRISPR/Cas9 technology. We found that *ASIP* knockout could down-regulate the expression of *PPARγ*, *FASN*, and *SCD*, thus affecting the saturation of fatty acids in milk and upregulating the expression of *FABP4*, *ELOVL6*, and *ACSL1,* increasing the synthesis of medium-long chain fatty acids. These results may allow the selection of potential targets in future molecular breeding efforts for dairy cows.

**Abstract:**

Agouti signalling protein (ASIP) is a coat colour-related protein and also is a protein-related to lipid metabolism, which had first been found in agoutis. According to our previous study, *ASIP* is a candidate gene that affects the lipid metabolism in bovine adipocytes. However, its effect on milk lipid has not been reported yet. This study focused on the effect of the *ASIP* gene on the lipid metabolism of mammary epithelial cells in cattle. The *ASIP* gene was knocked out in bMECs by using CRISPR/Cas9 technology. The result of transcriptome sequencing showed that the differentially expressed genes associated with lipid metabolism were mainly enriched in the fatty acids metabolism pathways. Furthermore, the contents of intracellular triglycerides were significantly increased (*p* < 0.05), and cholesterol tended to rise (*p* > 0.05) in bMECs with the knockout of the *ASIP* gene. Fatty acid assays showed a significant alteration in medium and long-chain fatty acid content. Saturated and polyunsaturated fatty acids were significantly up-regulated (*p* < 0.05), and monounsaturated fatty acids were significantly decreased in the *ASIP* knockout bMECs (*p* < 0.05). The Q-PCR analysis showed that knockout of *ASIP* resulted in a significant reduction of gene expressions like *PPARγ*, *FASN*, *SCD*, and a significant up-regulation of genes like *FABP4*, *ELOVL6*, *ACSL1*, *HACD4* prompted increased mid-to long-chain fatty acid synthesis. Overall, *ASIP* plays a pivotal role in regulating lipid metabolism in bMECs, which could further influence the component of lipid in milk.

## 1. Introduction

As one of the essential sources of nutrition for humans, milk contains several different components, such as protein, fats, and amino acids. Milk’s major energy component is fat. It contains several lipid fractions including triglycerides (over 95%), diacylglycerol (2%) free fatty acids (0.5%), cholesterol (0.5%), and phospholipids (1%) [1]. Research has found that high-fat milk may cause heart disease, weight gain, and obesity [2,3]. Tucker and colleagues found that Americans who habitually consumed high-at milk had shorter telomeres than those who drank low-fat milk [4]. The abundant fatty acids in milk could take responsibility for this adverse effect on humans, especially saturated and partially unsaturated fatty acids. Some studies have suggested that people need to reduce the intake of saturated fatty acids. Myristic acid (14:0) and palmitic acid (16:0) contribute to an increase in low-density lipoprotein (LDL) cholesterol in the blood, which increases the risk of cardiovascular disease [5]. Furthermore, linoleate (18:2N6) and linolenic (18:3N3) are the main polyunsaturated fatty acid in milk [5]. The 18:2N6 is a precursor for arachidonate (20:4N6), the main precursor for the synthesis of arachidate (20:0). Arachidate may enhance platelet aggregation, but it may also increase the risk of coronary artery disease [6]. On the other hand, oleic acid (C18:1N9), about 25% of milk lipid, has health benefits for humans. Studies showed that the abound of monounsaturated fatty acids (MUFA) could reduce the concentration of cholesterol, LDL-cholesterol and triacylglycerol in plasma [7]. Thereby, MUFA contributes to decreasing coronary artery disease risk [5]. The molecule mechanism of gene regulation on milk lipids has been paid more attention in the research area of milk quality. By interference of stearoyl-CoA desaturase 1 (SCD1) in bMECs, a positive correlation was found between *SCD1* expression and the amount of unsaturated fatty acid in bMECs [8]. Kadegowda et al. identified the peroxisome proliferator-activated receptor γ (PPARγ) as a key lipid synthesis pathway in bMECs and a target gene for fatty acid regulation of lipid metabolism [9]. Li et al. reported that single nucleotide polymorphisms (SNPs) within fatty acid synthase (FASN), peroxisome proliferator activated receptor gamma coactivator 1 alpha (PPARGC1A), ATP binding cassette subfamily G member 2 (ABCG2) and insulin like growth factor 1 (IGF1) were significantly associated with milk fatty acid composition in dairy cattle [10]. However, these regulatory mechanisms for the content and composition of milk lipid are still unclear. Therefore, more candidate genes involved in the regulation of milk lipid metabolism need to be explored.

Agouti signalling protein (ASIP) was first discovered and identified as a protein associated with hair colour in agoutis [11]. Another study on the *ASIP* gene reported that the mutation of *ASIP* in the promoter region is associated with the overexpression of *ASIP* in various tissues [12]. The *ASIP* mutant mice showed yellow coat colour, dramatically enhancing their weight, bulimic behaviour and diabetes mellitus [12]. Jones et al. found significant up-regulation of *FASN* and *SCD1* expression in a study of yellow agouti mutants [13]. They are involved in de novo fatty acid synthesis and desaturation in the liver and adipose tissue. This may lead to a significant increase in unsaturated fatty acids in medium-long chain fatty acids. Despite the ability of *ASIP* for the regulation fat deposition and lipid metabolism in adipose, no studies have focused on its effect on the synthesis and metabolism of milk lipid in cattle. Previous studies on *ASIP* in cattle have mainly focused on coat colour [11,12]. Recently, its function on bovine fat deposition was explored. Albrecht et al. found that Japanese black cattle with higher intramuscular fat content had higher ASIP mRNA levels in the longissimus dorsi muscle than Holstein cattle [14]. Liu et al. reported that *ASIP* mRNA expression levels in subcutaneous fat of 229 Bulls of the F2-generation of a Charolais × Holstein cross were significantly correlated with bovine fat deposition traits carcass fat weight [15]. Furthermore, Liu et al. screened three SNPs in *ASIP* genes in a Chinese Simmental cattle population. The results indicated that two SNPs correlated with carcass and fat-related traits significantly, such as live weight and backfat thickness [16]. These further suggest a significant association between *ASIP* and fat accumulation in cattle. Previous studies in our laboratory found that the recombinant ASIP protein could regulate the expression of adipogenic and fatty acid metabolism-related genes in adipocytes, such as leptin (LEP), adiponectin (ADIPOQ), solute carrier family 27 member 6 (SLC27A6), etc. Hence, *ASIP* could be a potential candidate gene involved in milk lipid metabolism.

To explore the effect of *ASIP* on bovine milk lipid, *ASIP* knock-out bMECs were constructed by CRISPR/Cas9 technology in this study. The triglycerides, cholesterol, and fatty acid contents in the knock-out cell lines were detected. Moreover, the transcriptome of *ASIP* knock-out bMECs was obtained and analysed by RNA-seq. The results of this study can reveal the potential effects of *ASIP* in bovine mammary epithelial cells on milk lipid metabolism. It could provide useful information about candidate genes for the future breeding of high-quality dairy cows.

## 2. Materials and Methods

### 2.1. Cell Culture

Lactating Chinese Holstein dairy cows were selected to isolate bovine mammary epithelial cells (bMECs). Fresh mammary tissue was taken and placed in a solution of PBS (Tiangen, Beijing, China) containing 5% (*v*/*v*) penicillin and streptomycin (Tiangen, Beijing, China). It was stored in a 37 °C thermostat and transported immediately to the laboratory. The tissue was washed several times with PBS containing antibiotics until the solution was no longer turbid. The bulk breast tissue was divided into small pieces, cut into 1 mm^3^, and washed with 1 PBS. Subsequently, the bMECs were isolated by the tissue block culture method [17]. These bMECs were preserved in the Laboratory of Molecular Genetics, College of Animal Sciences, Jilin University. According to our previous work, bMECs were cultured in DMEM/F12 (HyClone, Logan, UT, USA) supplemented with 10% foetal bovine serum (FBS, 11011–6123, Tian Hang, Zhejiang, China) at 37 °C and 5% CO_2_ in an incubator (Thermo Fisher Scientific, Waltham, MA, USA). All the experiments were performed according to the guidelines for the care and use of laboratory animals by Jilin University (Animal Care and Use Committee permit number: SY201901007) [18].

### 2.2. Construction of KO-ASIP bMECs

The target sequences of sgRNAs in the bovine *ASIP* exon 1 were predicted using Zhang Lab (https://zlab.bio/guide-design-resources (accessed on 3 June 2017)), and the details of the sgRNAs sequence were showed in Table 1. All the DNA Oligos were synthesized by Sangon (Shanghai, China). Each sgRNA was inserted into the PX459 vector (BioVector NTCC, Beijing, China) and named gASIP/PX459. Knockout of the *ASIP* gene fragment using transient transfecting technology in bMEC. After 24 h, the bMECs with *ASIP* gene knockout were screened using 2 μg/mL puromycin dihydrochloride (Beyotime, Jiangsu, China). Viable cells were cultured in 20% FBS and 1% penicillin-streptomycin solution. These cells were trypsinized and collected by centrifugation, resuspended in medium and diluted well. Cells were cultured in 96-well plates at one cell per well.

### 2.3. Identification of KO-ASIP bMECs Lines

The genomic DNA of bMECs was extracted following the Phenol-Chloroform Extraction protocol [19]. The PCR amplification was performed using primers to identify the knockout *ASIP* gene in bMECs. The forward primer and reverse primer sequences were 5′-CAGGTCCATCCTATCCTTCTG-3′ and 5′-CAGGTCCATCCTATCCTTCTG-3′, respectively. The PCR products were sent to GENEWIZ (Suzhou, China) for Sanger sequencing. The single-cell cloning of bMEC with *ASIP* gene-edited was screened, and the bMEC was named KO-*ASIP*.

### 2.4. Determination of Triglycerides, Cholesterol, and Intracellular Fatty Acid Content in bMECs of ASIP Gene

Cells were seeded at 6 × 10^6^ cells/well in 6-well culture plates with triplicates for KO-*ASIP* bMECs and WT bMECs. These bMECs were digested with 0.25% trypsin, the cell suspension was collected with a 1.5 mL clean centrifuge tube, and cells were collected by centrifugation at 700 r. It was washed again with PBS. Add Lysis Buffer for Triglyceride or Cholesterol Extraction. Triglycerides and CHOL were measured using a TG detection kit and a tissue/cell total CHOL assay kit (E1015-105 and E1013, Applygen Technologies, Beijing, China), respectively. The cellular contents of TG and CHOL were normalized by protein content. The concentrations were measured using a microplate reader (SM600, Shanghai YongChuang Medical Instrument Co., Ltd., Shanghai, China). Cells were collected at a cell volume of 50 mg per tube, three tubes each from KO-*ASIP* bMECs and WT bMECs and sent to Shanghai Applied Protein Technology Inc to determine the intracellular fatty acid content.

### 2.5. RNA Extraction, Library Preparation and RNA-Seq Analysis

Total RNA was extracted from the bMECs by TRIzol reagent (Invitrogen, Carlsbad, CA, USA). The RNA quality detection was carried out using the Agilent bioanalyzer 2100 (Agilent Technologies, Palo Alto, CA, USA). A total of 1 μg RNA with RIN value above 7 was used following library preparation. Next-generation sequencing library preparations were constructed according to the manufacturer’s protocol (NEBNext^®^ Ultra™ RNA Library Prep Kit for Illumina^®^). The poly(A) mRNA was captured with NEBNext^®^ Poly(A) mRNA Magnetic Isolation Module (NEB, Beijing, China). The mRNA fragmentation and priming were performed using NEBNext First Strand Synthesis Reaction Buffer and NEBNext Random Primers. First-strand cDNA was synthesized using ProtoScript II Reverse Transcriptase, and the second-strand cDNA was synthesized using Second Strand Synthesis Enzyme Mix. The RNA libraries were structured using cDNA NEBNext Ultra RNA Library Prep Kit for Illumina and purified by Beckman Agencourt AMPure XP beads (Beckman Kurt, Shanghai, China). Library detection and quantification were carried out using Agilent bioanalyzer 2100 and Qubit. TruSeq PE Cluster Kit V4 was used for cBOT automatic mounting. The HiSeq sequencing was implemented using TruSeq SBS Kit v4-HS. DEGs were analysed between KO-*ASIP* and WT bMECs using the DESeq Bioconductor package [20]. This analysis was based on the negative binomial distribution and was adjusted using Benjamini and Hochberg’s approach for controlling the false discovery rate. The fluorescent quantitative PCR validation assay was performed using |log2FC| ≥ 1.5 and *p* < 0.05 as screening conditions for differential gene expression, and 10 genes related to lipid metabolism and related marker genes were selected for quantitative validation. The gene ontology (GO) analysis of DEGs was conducted by the Cluster-Profiler R package [21], in which the gene length bias was corrected. The GO terms with corrected *p* < 0.05 were significantly enriched by DEGs. The KEGG is a database resource for understanding high-level functions and utility of the biological systems from molecular-level information (http://www.genome.jp/kegg/ (accessed on 6 February 2018)). We used the cluster Profiler R package software to test the statistical genes.

### 2.6. Determination of Gene Expression Related to Lipid Metabolism

To analyse gene expression of fatty acid metabolism in KO-*ASIP* cells, SYBR green-based Q-PCR was used. Initially, total RNA was extracted from 4 × 10^6^ cells using FastPure^®^ Cell/Tissue Total RNA Isolation kit (Vazyme, Nanjing, China) following the manufacturer’s instructions. The RNA was quantified by NANODROP 2000 spectrophotometry (Thermo, Shanghai, China). cDNA was prepared using the EasyScript^®^ All-in-One First-Strand cDNA Synthesis SuperMix for the qPCR kit (Trans, Beijing, China) following the manufacturer’s instructions. The cDNA obtained by reverse transcription of 1 μg of RNA was used as an RT-qPCR template using the Essential DNA Green Master Kit (Roche, Beijing, China) and PCR-Max-EC048 Real-Time PCR system (Bibby scientific, Staffordshire, UK). The RT-qPCR reactions were performed at 95 °C for 5 min, followed by 45 cycles of 95 °C for the 30 s, 60 °C for 30 s, and 72 for 20 s. A melting curve was run for each assay. All RT-qPCR reactions were performed in three biological replicates and three technical replicates. mRNA level of each gene was normalized to β-actin. The information on primers pairs was listed in Table 2. The expression fold change was calculated through a data analysis web portal using the ΔΔCt method, in which ΔCt is calculated between the gene of interest and an average of reference genes, followed by ΔΔCt calculations [ΔCt (test group) −ΔCt (control group)]. Fold change was then calculated using the 2^−ΔΔCt^ formula [22].

### 2.7. Statistical Analysis

Data from experiments are shown as mean ± standard error of the mean. Differential significance was determined by *p* < 0.05. Gene expression data from qRT-PCR experiments were analysed with the comparative Ct method (2^−ΔΔCt^). A two-tailed *t*-test was conducted on the data using GraphPad Prism 7.0 (GraphPad Software Inc., San Diego, CA, USA).

## 3. Results

### 3.1. Construction of KO-ASIP bMECs

In this study, the gASIP/pX459 re-combinant plasmid was successfully constructed by designing gRNA in the CDS promoter part of the ASIP gene (Figure 1a,b). After screening by stylomycin, the KO-*ASIP* bMECs were identified with PCR. Electrophoresis on an agarose gel (Figure 1c), identified that the PCR product had a length of 980 bp; after that, we sequenced this product. Comparing the KO-*ASIP* cell line sequence with the WT sequence revealed that the gRNA sequence had a deletion of 47 bases near the PAM site when the cell succeeded in transfecting the gRNA1 sequence carrier, resulting in shift mutation (Figure 1d). This result proved that the *ASIP* gene had been knock-outed successfully inside the bMEC. The result of qPCR found that *ASIP* expression was significantly decreased in the KO-*ASIP* bMEC line compared to the WT bMEC line (Figure 1e).

### 3.2. Detection of Triglycerides, Cholesterol and Fatty Acid in KO-ASIP bMECs

The results showed that the intracellular triglyceride content was increased (*p* < 0.05), and the cholesterol content was not significantly changed in the *ASIP* knockout bMECs (Figure 2a,b). Fatty acids were extracted from the wild-type and the *ASIP* gene knockout cells, respectively. The composition and content of the fatty acids were analysed. There were significant reductions in monounsaturated fatty acids (MUFA) and saturated fatty acids (SFA), and polyunsaturated fatty acids (PUFA) in KO-*ASIP* compared to WT. Both polyunsaturated fatty acids ω-3 (N3) and polyunsaturated fatty acids ω-6 (N6) were elevated significantly in KO-*ASIP* bMECs (*p* < 0.05, Figure 2c,d). Furthermore, a total of 39 fatty acids were detected. In *ASIP* knockout cells, the level of saturated fatty acids C14:0, C15:0, C16:0, C17:0, C18:0, C20:0, C21:0, C24:0 and polyunsaturated fatty acids such as C18:2N6, C18:3N6, C18:3N3, C20:2N6, C20:3N6, C20:4N6, C20:3N3, C20:5N3, C22:2N6, C22:5N3, C24:1N9 was significantly higher than that in WT. A significant increase of C16:1N7, C18:1TN9, C18:1N9, and C20:1N9 occurred in *ASIP* knockout bMECs. No significant differences were found in the other 16 fatty acids (*p* < 0.05, Figure 2e,f, Table 3).

### 3.3. Functional Enrichment Analysis of the DEGs in KO-ASIP bMECs

The RNA-seq analysis of the KO-*ASIP* bMECs was performed to elucidate how *ASIP* genes regulate lipid metabolism. A total of 485 differentially expressed genes (DEGs), including 174 up-regulated and 311 down-regulated genes, were identified in KO-*ASIP* bMECs compared with wild type (Figure 3a,b), which were also presented in the form of a heat map (Figure 3c). Examples: *RARRES2*, *FABP4*, *ELOVL6*, *HACD4* and *NR4A1*. All differential genes in the comparison group were selected and set for clustering analysis. The FPKM values of genes were clustered using hierarchical clustering, and genes or samples with similar expression patterns in the heat map were clustered together (Figure 3d,e). Ten differentially expressed genes (seven up-regulated differentially expressed genes and three down-regulated differentially expressed genes) were selected for qPCR analysis to verify the results of RNA-seq. The qPCR results showed that the relative expression trends of the 10 differentially expressed genes detected in WT and KO-*ASIP* bMECs were consistent with the RNA-seq results, confirming the accuracy of the RNA-seq data (Figure 4 and Table 4).

### 3.4. Prediction and Analysis of Signal Pathway Interaction after ASIP Knockout

They were significantly enriched in 28 GO terms (*p* < 0.05), including 22 biological processes, 3 cellular components, and 3 molecular function-related GO terms. The GO terms most significantly different in biological processes, cellular components, and molecular function categories are shown in Figure 3d and Table 5. The GO terms of the most important biological process include the immune system, single-organism, cellular, biological regulation, response to stimulus, metabolic, and multicellular organismal processes. The most significant cell components of the GO terms, include the cell part, membrane part, organelle, extracellular region part, and membrane. The most significant molecular function GO terms include binding, catalytic activity, structural molecule activity, signal transducer activity and transporter activity.

### 3.5. Verification of Differential Gene Expression Level

A KEGG enrichment analysis of differential expression genes showed that differential expression genes were significantly enriched into 46 KEGG pathways, with the first 30 most significant ones shown in Figure 3e, including the PPAR signalling pathway, Adipocytokine signalling pathway and AGE-RAGE signalling pathway in diabetic complications. Differentially expressed genes related to lipid metabolism are mainly enriched in fatty acid metabolism pathways, such as long-chain fatty-acyl-CoA biosynthetic process, long-chain fatty acid metabolic process, unsaturated fatty acid biosynthetic process et al. (Table 6). We validated the *SLC26A2*, *BGN*, *ACSL1*, *PGM2L1*, *SCD*, *RARRES2*, *FABP4*, *ELOVL6*, and *HACD4* and *NR4A1* genes in the differentially expressed genes by RT−qPCR, The quantitative results showed a significant upregulation of *SLC26A2*, *BGN*, *ACSL1*, *FABP4*, *ELOVL6*, and *NR4A1* (*p* < 0.05, Figure 4) when compared with WT bMECs, but not significantly for *HACD4*. *PGM2L1*, *SCD*, and *RARRES2* were significantly down-regulated (*p* < 0.05, Figure 4).

### 3.6. Effect of ASIP Knockout on the Milk Lipid Metabolism Pathways

A quantitative analysis of genes associated with the fatty acid metabolism pathway was performed using qPCR technology to resolve the regulatory effect of *ASIP* knockout on genes involved in the fatty acid metabolism pathway in bovine mammary epithelial cells. The results showed that the knockout of *ASIP* affected genes related to fat and fatty acid metabolism in bovine mammary epithelial cells. Knockout of *ASIP* resulted in a significant down-regulation of mRNA levels in key fatty acid metabolism genes such as *PPARγ*, *FASN*, and *SCD1* (*p* < 0.05, Figure 4). However, key genes like *FABP4*, *ELOVL6*, and *HACD4* associated with fatty acid transport and elongation were significantly up-regulated (*p* < 0.05, Figure 4).

## 4. Discussion

Previous work in our laboratory has demonstrated that the addition of recombinant protein *ASIP* altered mRNA expression of genes related to lipid metabolism and significant increases triglycerides and cholesterol content in bMECs (unpublished data). These studies could imply the involvement of *ASIP* genes in the synthesis and catabolism of lipids in cattle. To further explore the effect of the *ASIP* gene on lipid metabolism in bMECs, we successfully constructed the *ASIP* knockout bMECs using CRISPR/Cas9 technology. The results showed that the expression of *ASIP* was significantly reduced at the mRNA level in the knockout cell lines.

The content of triglycerides, cholesterol and fatty acids was firstly measured to investigate further the effect of *ASIP* knockout on lipid metabolism in bMECs. The results presented that *ASIP* knockout affected the composition of medium and long-chain fatty acids, increasing saturated and polyunsaturated fatty acids content and a decreased level of monounsaturated fatty acids in bMECs. The content of saturated fatty acids myristic acid (C14:0), pentadecanoic acid (C15:0), palmitic acid (C16:0), heptadecanoic acid (C17:0), stearic acid (C18:0), arachidonic acid (C20:0), 21-carbonic acid (C21:0), and ditetradecanoic acid (C24:0) were significantly up-regulated. The significant increase in saturated fatty acids of 14-24 C-atoms, which are substrates for triglycerides production [23], could provide the fundamental element for the elevation of triglycerides in bMECs after knockout of the *ASIP* gene. It has been widely accepted that saturated fatty acids and TG in dairy fat, which may lead to obesity, and high blood pressure, are detrimental to health. Moreover, saturated fatty acids such as myristic (14:0) and palmitic (16:0) acids could enhance the level of low-density lipoprotein (LDL- and high-density lipoprotein (HDL-) cholesterol in the blood [24]. Despite their inherent ability to inhibit bacteria and viruses, a high intake of these fatty acids could raise blood cholesterol content which could cause heart disease, weight gain, and obesity [5]. For some polyunsaturated fatty acids, there were significant increases in linoleic acid (C18:2N6), gamma-linoleic acid (C18:3N6), and alpha-linolenic acid (C18:3N3) in bMECs with *ASIP* knockout. The elevation of these fatty acids could lower blood lipids and improve vascular health status [25,26]. Ni Dan et al. investigated the effect of the complete absence of certain long-chain fatty acids (LCFA) on milk lipid metabolism in bMECs using the univariate principle. It was found that triglycerides synthesis was significantly down-regulated in bMECs with the absence of C18:0, C18:2N6 or C18:3N3 [27]. This implies that elevated C18:2N6 and C18:3N3 may also have a facilitating effect on triglycerides synthesis. Furthermore, the monounsaturated fatty acid oleic acid (C18:1N9), which accounted for the largest proportion of milk lipid [28], and other monounsaturated fatty acids such as palmitoleic acid (C16:1N7) and C20:1N9 were significantly down-regulated under *ASIP* knockout conditions. It has been indicated that consuming diets containing high amounts of monounsaturated fatty acids effectively reduces plasma lipids, such as cholesterol and triacylglycerol concentrations [29]. Furthermore, monounsaturated fatty acids are more protective against atherosclerosis than polyunsaturated fatty acids [30]. Thus, the reduction of *ASIP* can alter fatty acid composition in the intracellular bMECs and may further in milk. However, this alteration in milk may be detrimental to human health. It suggested that the lower expression level of *ASIP* in mammary epithelial cells was not recommended to breed dairy cows with high milk quality in the future.

To preliminarily investigate the molecular mechanism of the effect of *ASIP* on the lipid metabolism in bMECs, the transcriptome sequencing was analysed. The results showed that 485 differentially expressed genes were detected, including 311 up-regulated genes and 174 down-regulated genes. The GO terms analysis of differentially expressed genes indicated 112 differentially expressed genes were significantly enriched in 28 GO entries (*p* < 0.05), including 22 biological processes, three cellular components, and three molecular functionally relevant GO entries. The KEGG analysis indicated significant enrichment of differential expressed genes into 46 KEGG pathways, including PPAR signalling pathway, Adipocytokine signalling pathway and AGE-RAGE signalling pathway in diabetic complications. We screened 10 differentially expressed genes to satisfy the conditions of |log2FC| ≥ 1.5 and *p* < 0.05 as significant differential expression. The quantitative results showed that although our transcriptome was not biologically replicated, the sequencing results and relative gene quantification results were relatively consistent. Differentially expressed genes related to lipid metabolism are mainly enriched in fatty acid metabolism pathways, such as long-chain fatty-cyl-CoA biosynthetic process, long-chain fatty acid metabolic process, unsaturated fatty acid biosynthetic process, etc. The qPCR results of differentially expressed genes indicated that expression of *ACSL1*, *FABP4* and *ELOVL6* genes was elevated and the *SCD FASN* expression was decreased in the knockout cell lines, consistent with the transcriptome sequencing results.

Key genes of the fatty acid metabolic pathway were detected to explore the molecular mechanisms of fatty acid alterations in bMECs after *ASIP* knockout. As an enzyme directly related to lipid synthesis and fatty acid transport and degradation, ACSL1 can significantly influence fatty acid metabolism. By overexpressing *ACSL1* in bovine adipocytes, Zhao et al. found a significant increase in saturated and polyunsaturated fatty acid (PUFA) content, particularly C16:0 and C18:0. Overexpression of ACSL1 further increased the proportion of eicosapentaenoic acid (EPA) [31]. In the present study, *ACSL1* expression was significantly up-regulated in *ASIP* knockout bMEC, which may lead to SFA accumulation such as palmitic acid (C16:0) and C18:0 content as well as an increase in PUFA and C20:5N3. Moreover, the primary function of *FABP4* is to transport fatty acids across the membrane, and overexpression increases FA transport to enhance energy and lipid metabolism [32]. The deletion of *FABP4* results in an impaired fatty acid efflux, leading to an increase in fatty acids in FABP4-deficient cells. It has been shown that FABP4 maintains eicosanoid homeostasis in macrophages in mice [33]. The FASN gene is a multifunctional peptidase for producing saturated fatty acids. It is responsible for all steps in the ab initio synthesis of palmitic acid (C16:0) from acetyl-coenzyme an (acetyl coenzyme a) and malonyl coenzyme a (malonyl coenzyme a) [34]. The specific knockout of *FASN* in mouse mammary glands significantly reduces the content of medium and total long-chain fatty acids in milk [35]. This may indicate the down-regulation of *FASN* may lead to a decreased synthesis rate of C16:0 and finally lower the level of C16:0 in bMEC with *ASIP* knockout. Furthermore, *SCD* is the rate-limiting enzyme that catalyses the synthesis of monounsaturated fatty acids (MUFAs) from saturated fatty acids (SFAs). The major substrates of *SCD* are C16:0 and C18:0 FA, which can be converted to C16:1 cis9 and C18:1 N9 [36]. Association analysis of SNP loci of *SCD* with milk lipid ty acid traits in 297 Holstein cows was conducted by Mele et al. They found that cows with the AA genotype had higher levels of cis-9C18:1, total monounsaturated fatty acid levels and C14:1/C14:0 ratios in milk compared to cows with the VV genotype [37]. In this study, there was a significant down-regulation of *SCD* expression in *ASIP* knockout bMEC, leading to SFA accumulation, such as palmitic acid (C16:0) and C18:0 content and the reduction of MUFA, like C16:1N7 and C18:1N9. For C16:0, both *FASN* and *SCD* could influence its level in bMEC. As C16:0 content was lower in the control group, it may imply that the consumption rate of C16:0 by *SCD* could be lower than the synthesis rate by *FASN* in *ASIP* knockout cells. Furthermore, Junjvlieke et al. used adenovirus to overexpress the *ELOVL6* gene, a rate-limiting enzyme in the long-chain fatty acid elongation reaction, in bovine adipocytes and found a significant increase in the proportion of C18:0 and C20:4N6 fatty acids [38]. Our results suggested that the mRNA expression level of *ELOVL6* was significantly increased in bMEC with *ASIP* knockout which could contribute to the upregulation of C18:0. Additionally, the expression of *HACD4* mRNA was detected in this study, as it is an important enzyme at the third step of fatty acid extension. Even though there was a significant elevation in 18:2N6 and 18:3N3 fatty acids after *ASIP* knockout in bMECs, the higher mRNA expression of *HACD4* was not detected. It may indicate that the enhancement of 18:2N6 and 18:3N3 fatty acids in *ASIP* knockout of bMECs was not mainly regulated by *HACD4*. Moreover, we found that the knockout of the *ASIP* gene in bMECs resulted in significant downregulation of the peroxisome proliferator-activated receptor (PPARγ), which has a key role in lipid metabolism, promoting free fatty acid uptake and accumulation of triacylglycerols in adipose tissue and liver [39]. The down-regulation of *PPARγ* may lead to triglycerides down-regulation. Garin-Shkolnik et al. found that high expression of fatty acid-binding protein 4 (FABP4, also known as aP2) could down-regulate *PPARγ* expression [40]. Tingting Li et al. found that in human hepatocytes overexpressing *ASCL1* caused a significant decrease in fatty acid synthesis pathway-related genes such as *FASN* and *SCD1* by suppressing *PPARγ* expression to an increase in triglycerides levels [41]. This is consistent with the results of gene and triglycerides changes in our experiments. In another human hepatocyte line (L02 cells) overexpressing *PPARγ*, it was found that up-regulation of PPARγ could affect cholesterol efflux and thus lower cholesterol levels through multiple pathways [42]. Our cholesterol assay results indicate a trend of up-regulation, but one not significant in bMECs after knockout of *ASIP*, but its molecular mechanism needs to be further investigated. These results suggest that the knockout of *ASIP* can influence the expression of key genes in the fatty acid metabolism pathways, thus affecting the intracellular lipid metabolism, especially the composition of fatty acids. It has been shown that *ASIP* affects the cellular content of cAMP by competitive binding to MC1R [43,44]. However, whether *ASIP* affects the expression of key genes in lipid metabolism in bMECs by regulating cAMP content still needs further investigation.

## 5. Conclusions

In summary, the knockout of *ASIP* by using Cas9 technology could alter lipid composition in bMECs, such as the increased content of triglycerides, saturated fatty acids, and polyunsaturated fatty acids. Some of the differentially expressed genes that were screened by transcriptome sequencing and related to fatty acid synthesis pathway were then analysed to preliminarily elucidate the regulatory mechanism of the *ASIP* gene in fatty acid metabolism in bMECs. These results could provide a reference for selecting potential targets in the molecular breeding work of dairy cattle in the future.

## Figures and Tables

**Figure 1 animals-12-01389-f001:**
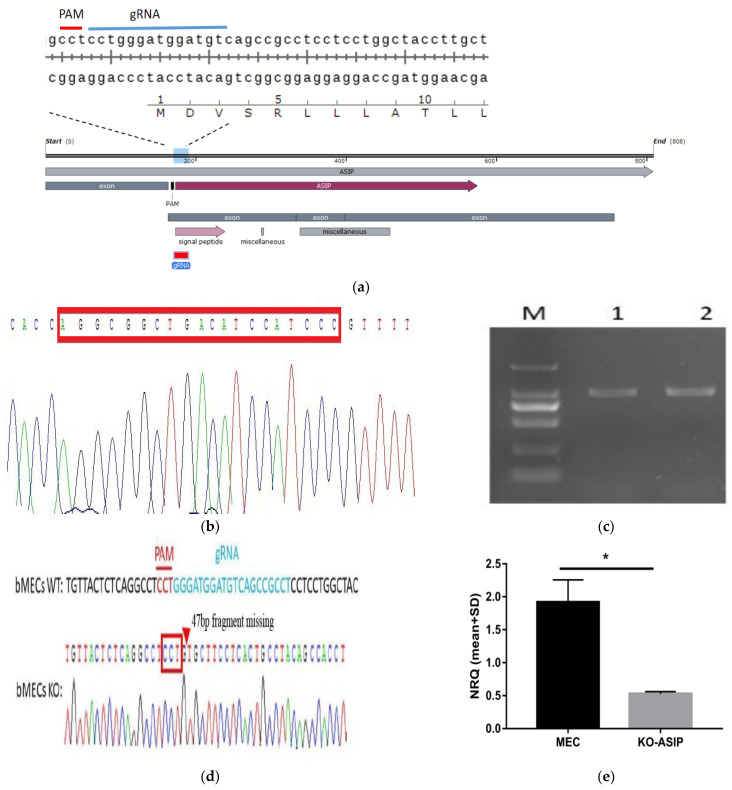
Construction of *ASIP* bMECs line: (**a**) gRNA’s location in the CDS region; (**b**) Sequencing map of positive bacterial liquid constructed by gASIP/PX459 vector; (**c**) *ASIP* knockout PCR validation of KO-*ASIP* MECs (M: market, 1: WT MEC, 2: KO-*ASIP* MEC); (**d**) Sequencing results of cell *ASIP* PCR products; (**e**) mRNA level validation of *ASIP* knockout (Ko-*ASIP* is an *ASIP* knockout bMECs. MEC is a wild-type bMECs. Error bars indicate SEM. * *p* < 0.05).

**Figure 2 animals-12-01389-f002:**
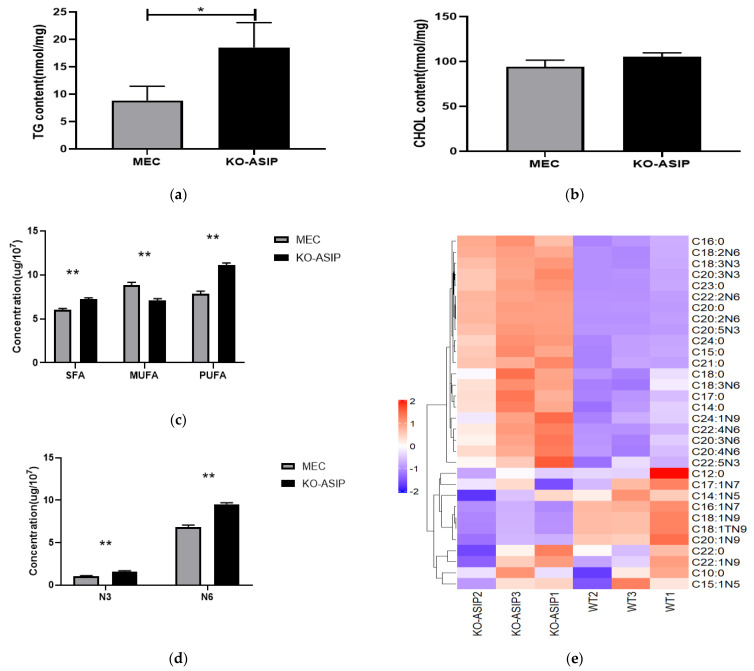
The effect of *ASIP* on TGs, CHOL and FA contents in bMECs: (**a**) The TG content in the cell; (**b**) The CHOL content in the cell; (**c**) The total content of various fatty acids in a cell; (**d**) The total content of N3 and N6 fat acids in a cell. Error bars indicate SEM; (**e**,**f**) Any of various medium and long-chain fatty acids in the cell. Error bars indicate SEM. * *p* < 0.05, ** *p* < 0.01, ns: *p* > 0.05.

**Figure 3 animals-12-01389-f003:**
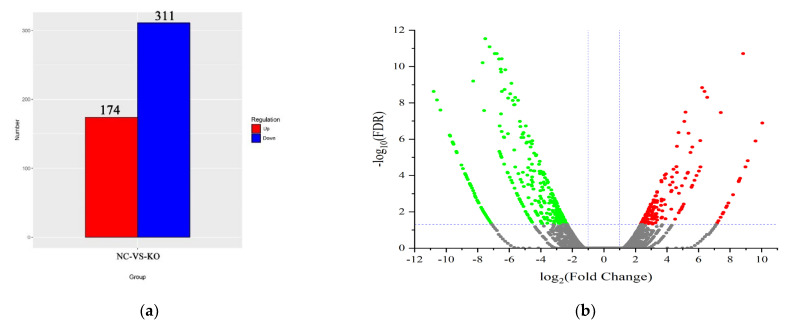
Transcriptome sequencing analysis after *ASIP* knockout: (**a**) Number of genes differentially expressed by KO-*ASIP* and NC; (**b**) Volcano map of differential genes, red dots of significantly different genes indicate up-regulation, green dots indicate down-regulation; the abscissa represents the change of gene expression multiple in different samples. The ordinate represents the statistical significance of changes in gene expression levels; (**c**) Differential gene cluster plots were clustered with log10 (rpkm1) values, red for high expressed genes and blue for low expressed genes. Colours range from red to blue, indicating higher gene expression; (**d**) Rich distribution map of GO Term; (**e**) Rich distribution map of KEGG gene (KO is KO-*ASIP* bMECs and NC is WT bMECs).

**Figure 4 animals-12-01389-f004:**
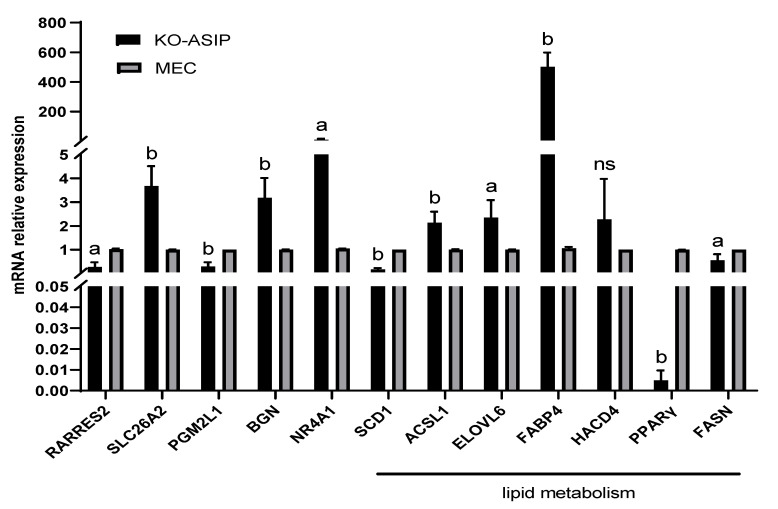
Quantitative results of genes related to fatty acid metabolism pathway. Error bars indicate SEM. a: *p* < 0.05, b: *p* < 0.01, ns: *p* > 0.05.

**Table 1 animals-12-01389-t001:** Agouti signalling protein (ASIP) gene gRNA Sequence and Primer.

Name	Sequence (5′ -> 3′)
ASIP gRNA	CCT GGGATGGATGTCAGCCGCCT
ASIP gRNA 1F	CACCAGGCGGCTGACATCCATCCCAGG
ASIP gRNA 1R	AAACGGGATGGATGTCAGCCGCCT

**Table 2 animals-12-01389-t002:** RT-qPCR Primer.

Gene Name	Sequence (5′ -> 3′)		Length (bp)	Tm (°C)
*Agouti signalling protein* (ASIP)	Forward primer	CAGGTCCATCCTATCCTTCTG	21	56.36
	Reverse primer	CACCAAGTGCCTTGACTTTG	20	54.81
*RARRES2*	Forward primer	GAAGAAAGACTGGAGGAAAGA	21	54.88
	Reverse primer	CGTTGAACCTGAGTCTGTATG	21	56.39
*ELOVL6*	Forward primer	TCGAACTGGTGCTTATATGG	20	54.96
	Reverse primer	TGTATCTCCTAGTTCGGGTG	20	55.78
*PGM2L1*	Forward primer	GCTTTGTAGTTGGGTATGAC	20	54.01
	Reverse primer	GATACACAGGAACATCTTTGG	21	54.29
*HACD4*	Forward primer	CCAAGAGAAATACGTGGTGT	20	55.4
	Reverse primer	GATAAATTGGCATCCACAGG	20	54.39
*FABP4*	Forward primer	TGAAATCACTCCAGATGACAG	21	55.33
	Reverse primer	CATTCCAGCACCATCTTATC	20	53.83
*SLC26A2*	Forward primer	CCCAATCCATCGCTTATTCT	20	55.27
	Reverse primer	CACCAATCATAAGGCACAGT	20	55.72
*BGN*	Forward primer	TGGACCTGCAGAATAATGAC	20	55.13
	Reverse primer	CTTGGAGATCTTGTTGTTCAC	21	54.8
*NR4A1*	Forward primer	TGCTTCCTTCAAGTTCGAG	19	55.17
	Reverse primer	GACTGCCATAGTAGTCAGAG	20	54.42
*FASN*	Forward primer	TGGAGTACGTTGAAGCCCAT	20	59.02
	Reverse primer	ACTTGGTGGACCCAATCCG	19	59.62
*ACSL1*	Forward primer	TATACGAAGGTTTCCAGAGG	20	53.89
	Reverse primer	CTGCCATATCTTCAACCTGT	20	55.12
*PPARγ*	Forward primer	ACCCGATGGTTGCAGATTAT	20	56.97
	Reverse primer	CTTACTGTACAGCTGAGTCTT	21	54.73
*SCD1*	Forward primer	TTACACTTGGGAGCCCTAT	19	54.92
	Reverse primer	CTTTGTAGGTTCGGTGACTC	20	55.82

*ASIP*: Agouti signalling protein; *RARRES2*: Retinoic acid receptor responder protein 2; *ELOVL6*: ELOVL fatty acid elongase 6; *PGM2L1*: Phosphoglucomutase 2 like 1; *HACD4*: 3-hydroxyacyl-CoA dehydratase 4; *FABP4*: Fatty acid-binding protein 4; *SLC26A2*: Solute carrier family 26 member 2; *BGN*: Biglycan; NR4A1: Nuclear receptor subfamily 4 group A member 1; *FASN*: Fatty acid synthase; *ACSL1*: acyl-CoA synthetase long chain family member 1; *PPARγ*: Peroxisome proliferator activated receptor gamma; SCD1: Stearoyl-CoA desaturase 1.

**Table 3 animals-12-01389-t003:** Fatty acids with significant differences in content.

Name	Fatty Acids	ASIP/(μg/10^7^)	WT (μg/10^7^)	*p*-Value
C22:2N6	cis-13,16-Docosadienoic acid ester	0.014838855	0.003033941	2.58062 × 10^−5^
C20:2N6	cis-11,14-Eicosadienoic acid ester	0.341624903	0.145442912	3.0034 × 10^−5^
C20:0	Arachidate	0.307623681	0.120142089	3.37505 × 10^−5^
C18:2N6	Linoleate	7.768339337	5.578928386	7.88289 × 10^−5^
C20:5N3	cis-5,8,11,14,17-Eicosapentaenoic acid ester	0.104328937	0.0401591	1.18454 × 10^−4^
C18:3N3	Linolenate	0.62363992	0.425347349	1.99928 × 10^−4^
C16:1N7	Palmitoleate	0.164777719	0.261128788	4.0848 × 10^−4^
C16:0	Palmitate	4.350355862	3.616156662	4.52232 × 10^−4^
C23:0	Tricosanoate	0.005332711	0.002470432	5.41577 × 10^−4^
C20:3N3	cis-11,14,17-Eicosatrienoic acid ester	0.187933366	0.112294234	6.59976 × 10^−4^
C21:0	Heneicosanoate	0.011796608	0.005044538	8.50669 × 10^−4^
C15:0	Pentadecanoate	0.070234556	0.053131791	1.005741 × 10^−3^
C18:1N9	Oleate	3.148316962	3.978008211	1.364646 × 10^−3^
C18:1TN9	Elaidate	3.069077829	3.825630862	1.423762 × 10^−3^
C24:0	Tetracosanoate	0.097692958	0.044705298	1.431508 × 10^−3^
C20:4N6	Arachidonate	0.744710508	0.580317797	5.513477 × 10^−3^
C17:0	Heptadecanoate	0.110713207	0.095970131	6.445856 × 10^−3^
C22:4N6	Docosatetraenoate	0.107525698	0.07236041	7.025267 × 10^−3^
C20:1N9	cis-11-Eicosenoic acid ester	0.297438676	0.347778714	8.552879 × 10^−3^
C14:0	Myristate	0.213007746	0.175155542	9.617887 × 10^−3^
C18:3N6	γ-linolenate	0.014223628	0.007856659	1.6396827 × 10^−2^
C20:3N6	cis-8,11,14-Eicosatrienoic acid ester	0.525239379	0.414319337	1.6616937 × 10^−2^
C18:0	Stearate	1.693107169	1.562829639	3.7789469 × 10^−2^
C22:5N3	Docosapentaenoate	0.646932832	0.481578127	4.7233379 × 10^−2^
C24:1N9	cis-15-tetracosenoate	0.105985694	0.090770742	4.8230263 × 10^−2^

**Table 4 animals-12-01389-t004:** Differentially expressed mRNAs in RNA-seq.

Gene	Log2FC	*p*-Value
*FABP4*	6.533276472	4.17 × 10^−3^
*HACD4*	1.890805053	5.67 × 10^−3^
*ELOVL6*	1.528882028	2.37 × 10^−2^
*NR4A1*	2.351458317	1.91 × 10^−2^
*SLC26A2*	2.516902391	2.94 × 10^−4^
*BGN*	7.414908474	3.28 × 10^−11^
*ACSL1*	1.487425813	3.23 × 10^−2^
*RARRES2*	−8.204826354	5.19 × 10^−6^
*PGM2L1*	−2.243074323	3.13 × 10^−3^
*SCD*	−1.535957561	2.16 × 10^−2^

**Table 5 animals-12-01389-t005:** GO analysis results of genes related to lipid metabolism differences.

Term	Database	ID	Input Number	Background Number	*p*-Value	Corrected *p*-Value	Input
long-chain fatty-acyl-CoA biosynthetic process	Gene Ontology	GO:0035338	3	9	0.020237634	0.136990166	*ELOVL6*|*ACSL1*|*ACSL5*
very long-chain fatty acid biosynthetic process	Gene Ontology	GO:0042761	2	12	0.156216785	0.378399061	*ELOVL6*|*HACD4*
unsaturated fatty acid biosynthetic process	Gene Ontology	GO:0006636	2	12	0.156216785	0.378399061	*ELOVL6*|*SCD*
long-chain fatty acid transport	Gene Ontology	GO:0015909	1	7	0.340296604	0.500036513	*FABP4*

**Table 6 animals-12-01389-t006:** KEGG analysis results of genes related to lipid metabolism differences.

Term	Database	ID	Input Number	Background Number	*p*-Value	Corrected *p*-Value	Input
Fatty acid metabolism	Reactome	R-BTA-8978868	5	92	0.176804584	0.403842589	*FADS1*|*PTGES*|*ALOX12B*|*ALOX12*|*ACSL5*
Fatty acid metabolism	KEGG PATHWAY	bta01212	2	58	0.552242024	0.626722575	*FADS1*|*ACSL5*
Fatty acid biosynthesis	KEGG PATHWAY	bta00061	1	18	0.446287647	0.549649992	*ACSL5*

## Data Availability

Not applicable.

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
