# Peer review of "The Knockout of the ASIP Gene Altered the Lipid Composition in Bovine Mammary Epithelial Cells via the Expression of Genes in the Lipid Metabolism Pathway"

_animals, 2022, doi:10.3390/ani12111389_

Round 1
Reviewer 1 Report
Comments to authors:
Xie et al. investigate how lipid amounts and gene regulation changes in ASIP knockdown cells. This study can add relevant data for this bMEC cell line and generate new hypothesis of lipid metabolism relative to ASIP expression. If the ASIP KO is confirmed the dataset is certainly relevant but data representation still has to be significantly improved.
Therefore, I suggest the authors work on the following improvements:
- KO vs knockdown:
At this point there is a strong mRNA reduction level in the ASIP cell line(or cell lines?). However, the evidence provided could still be compatible with only a heterozygous deletion or maybe a mix of clones. The authors have to clarify this point. It is unclear if a pool of clones is analyzed or not.
- Figure 1:
It would be more useful to show where the gRNA is on the mRNA in panel a. and then where a STOP codon would be introduced in the protein after the 47bp deletion.
-b) the red sequence does not correspond exactly to the gRNA seq from Table 1. Why?
-c) this PCR verification does not show a visible difference on the gel. Use different primers and/or run the gel longer to show the difference. One needs to be able to differentiate between homo/heterozygous deletion. Was there a single clone generated or different ones?
This is also relevant for the repeat analyses. are the repeats technical replicates, or biological ones? With the same ASIP clone or different ones?
-Figure 2:
-Use same colors for MEC and KO-ASIP throughout figure and paper.
-C and D the labels cannot be read; SFA and PUFA abbreviations are not described or only in the discussion. Please mention earlier.
Figure 3:
- how and why were the 10 genes selected? I guess it comes in the discussion a bit but in the results there is no mention.
- all genes should be labelled in the volcano plot.
- I don’t understand the clustering analysis. The clustering was done according to how much the genes were misexpressed? I don’t see how this is relevant. What would the meaning of the colored groups be?
table 4 and Figure 4 should both be displayed in the same format for better comparison. For example both as log2FC and in the same direction. Why are the lists not congruent (10 vs 12 genes)?
- Table 5 and 6 list mainly non-significant results according to p-value. So what should we take from this?
Was RT PCR done in duplicates as mentioned in methods or more as mentioned in Figure legends?
- English has to be improved throughout the manuscript.
Just a few examples:
Minor:
- line 19: in vitro model for cells
- 21: ASIP knockdown? Should be knockout? See above. Be consistant.
-27: ever should be never; why spiny guiney pigs at this point??
-64: amount
-55: the intake of67: molecular mechanism
79: remove the
252: why were these genes chosen in the list.
270: all results are reversed? Why was this done like that?
310: table says HacD4 is NS but here it says it is significant.
440: alteration should read alter
442: differentially expressed genes
Author Response
Dear Reviewers
On behalf of all the authors of the submitted manuscript “The knockout of ASIP gene altered the lipid composition in bovine mammary epithelial cells via the expression of genes in lipid metabolism pathway”, we thank you so much for your kind reviews and meaningful suggestions. We have studied comments carefully and have made corrections which should meet with approvals we hope. The main revision in the paper (highlight changes in red) and the responses to the reviewers’ comments are as following:
Responds to Reviewer #1:
1.KO vs knockdown: At this point there is a strong mRNA reduction level in the ASIP cell line(or cell lines?). However, the evidence provided could still be compatible with only a heterozygous deletion or maybe a mix of clones. The authors have to clarify this point. It is unclear if a pool of clones is analyzed or not.
Response: Thank you very much for your suggestion. After the cell lines we used were screened for drugs, the cells were collected and fully diluted, and then re-seeded in 96-well plates with the number of one positive cell per well. Subsequent sequencing results showed that the KO-ASIP bMECs had successfully knocked out the 47 bp sequence including the ASIP gene promoter, so the cells were not mixed cells, and it was more likely that after the deletion of the ASIP promoter, there were other The coding method, or the transcript, resulted in the presence of ASIP expression in subsequent qRT-PCR assays.(line134-135)
- Figure 1: It would be more useful to show where the gRNA is on the mRNA in panel a. and then where a STOP codon would be introduced in the protein after the 47bp deletion.
Response: Thank you very much for your suggestion. We have modified Figure 1a, and the gRNA we designed finally resulted in the deletion in the first exon region of ASIP, which contains the promoter of ASIP. The deletion results in a frameshift mutation in the gene, which changes the gene and prevents transcription initiation.
- Figure 1b: The red sequence does not correspond exactly to the gRNA seq from Table 1. Why?
Response: Thank you very much for your suggestion. When designing the gRNA, PAM is not part of the gRNA, the gRNA sequence we designed with the website is the PAM-gRNA sequence: CCT GGGATGGATGTCAGCCGCCT, where CCT is the PAM sequence. When looking for the gRNA synthesized by the company, there is no PAM sequence in it. In order to be able to connect to the PX459 vector, when we synthesize the positive and negative strands of the gRNA, we need to add a CACC in front of the oligo sequence on the sense strand. The oligo sequence on the negative-sense strand is preceded by AAAC. These two sequences are complementary to the vector digested by BbsI, and the red part in Figure b is the ASIP gRNA positive-strand sequence obtained by sequencing, indicating that the gRNA was successfully integrated into the vector. We found that the picture b used before is the sequence of another gRNA integrated into the vector, we have replaced the correct picture b in the text, thank you for pointing out our problem in time.
- Figure 1c: this PCR verification does not show a visible difference on the gel. Use different primers and/or run the gel longer to show the difference. One needs to be able to differentiate between homo/heterozygous deletion. Was there a single clone generated or different ones?
Response: Thank you very much for your suggestion. Because the length of the increased gRNA is not long, only the fragment length of 47 bp is knocked out, so it is difficult to show obvious differences in gel electrophoresis. We had previously thought that the two could not be distinguished, but we have clearly obtained KO-ASIP bMECs after the subsequent sample delivery and sequencing, so we did not conduct further experimental design to specifically distinguish the two on the gel. Thank you for pointing out our shortcomings, we will further improve our experimental protocol when encountering such situations in the future. Unfortunately, due to time constraints, a solution to this problem cannot be provided here. In addition, the cells used here are not monoclonal cells, we used puromycin after plasmid introduction into recipient cells to screen a group of bMECs containing gRNA plasmids in the same cell culture dish. The surviving positive cells were not specifically cultured as single cells, but were used for group inoculation expansion and untreated WT bMECs cells inoculated in the same batch in units of culture wells.
- This is also relevant for the repeat analyses. are the repeats technical replicates, or biological ones? With the same ASIP clone or different ones?
Response: Thank you very much for your suggestion. In this part of the experiments, we used biological replicates. Subsequent experiments were performed using the same ASIP clone.
- Figure 2:Use same colors for MEC and KO-ASIP throughout figure and paper.
Response: Thank you very much for your suggestion. This is a negligence in our work. We have remade the pictures in the text and unified the colors of each group of pictures in Figure 2c, 2d, 2f.
- Figure 2C and D: the labels cannot be read; SFA and PUFA abbreviations are not described or only in the discussion. Please mention earlier.
Response: Thank you very much for your suggestion. We have marked the first occurrence of abbreviations such as SFA, PUFA, MUFA, N3, and N9 in the text, in line 280-282.
- Figure 3:how and why were the 10 genes selected? I guess it comes in the discussion a bit but in the results there is no mention.
Response: Thank you very much for your suggestion. Our transcriptome data has only one set of data, and no biological replicates were performed to verify the accuracy of the data. The genes with |log2FC|≥1.5 and P<0.05 were selected from the gene data obtained in the transcriptome sequencing report, and genes related to fat metabolism were selected as far as possible. This is to make up for the lack of repeated experiments in the transcriptome, and to further prove that this set of transcriptome data is in high consistency with the actual regulation of cellular gene expression. (Line 179 and Line 416-419)
- Figure 3b: all genes should be labelled in the volcano plot.
Response: Thank you very much for your suggestion. Transcriptomic analysis was screened based on |log2FC|, P-value, and FDR (false positive rate), and the range of P values was determined by controlling the FDR value. Relative to Bonferroni, FDR corrected p-values in a more modest way. It tries to achieve a balance between false positives and false negatives, and control the ratio of false/true positives to a certain range. The decision formula of FDR is: p * (n/i), p is the p-value of this test, n is the number of tests, i is the sorted position ID (for example, the i value of the largest P value must be n, the second The largest is n-1, and the smallest is 1). Only one set of transcriptome assays was performed in this experiment, and corresponding biological replicates were not performed. Most of the detected genes related to fat metabolism have a larger FDR than 0.05, which belongs to the possible false-positive range. Therefore, although the P-value obtained in transcriptomics is less than 0.05, because FDR is not in the screened range, the 10 verified in this experiment Among the genes, 7 are not in the significant part of the volcano plot, and only 3 are in the significant part, so the genes are not indicated here. In order to verify whether these genes with P≤0.05 and FDR greater than 0.05 in the transcriptome are credible, we subsequently selected 7 genes with large FDR for quantitative experiments in the process of verifying the feasibility of the transcriptome. The FDR value in the data is too large, but with the P-value as a reference, the expression of related genes in the transcriptome is still consistent.
- I don’t understand the clustering analysis. The clustering was done according to how much the genes were misexpressed? I don’t see how this is relevant. What would the meaning of the colored groups be?
Response: Thank you very much for your suggestion. This figure is a heat map based on the gene expression in each sample. In the figure, each column represents a sample, each row represents a gene, and the depth of the color in the figure represents the gene expression in the sample. In clustering, the closer the branch is, the closer the change law of its expression is. The genes with the same clustering pattern may have the same or related functions. (Line 302)
- Table 4 and Figure 4 should both be displayed in the same format for better comparison. For example both as log2FC and in the same direction. Why are the lists not congruent (10 vs 12 genes)?
Response: Thank you very much for your suggestion. Some genes are not based on differential genes obtained by transcriptome sequencing analysis, but are quantitatively derived from several key regulatory genes of fat metabolisms, such as PPARγ, based on related traits. Therefore, the results in Table 4 and Figure 4 are inconsistent. The table marked the possible differentially expressed genes screened under the conditions of |log2FC|≥1.5 and P<0.05.
- Table 5 and 6 list mainly non-significant results according to p-value. So what should we take from this?
Response: Thank you very much for your suggestion. One of the more interesting results we found in the transcriptome data was that ASIP knockout in breast cells had fewer genes that were altered in the fat metabolism pathway, but these genes were all related to fatty acid metabolism. Although the degree of enrichment was not significant in our transcriptome analysis, these lists are representative of lipid metabolism pathways. And it is not difficult to see that the effects are relatively consistent. This result, combined with the fat change trend we measured earlier, can clearly analyze the effect of ASIP on fatty acid metabolism.
- 13. Was RT PCR done in duplicates as mentioned in methods or more as mentioned in Figure legends?
Response: Thank you very much for your suggestion. We've made modifications to the unity, in fact, we've run out of the transcriptomics experiment, the other experiments have been done with 3 biological replicates and 3 control replicates.(Line197)
- English has to be improved throughout the manuscript.
Response: Thank you very much for your suggestion. We have made improvements to the language in the text and marked it in red.
- line 19: in vitro model for cells
Response: Thank you very much for your suggestion. We have done the modification. (Line 20)
- line 21: ASIP knockdown? Should be knockout? See above. Be consistant.
Response: Thank you very much for your suggestion. We have done the modification.
- line 27: ever should be never; why spiny guiney pigs at this point??
Response: Thank you very much for your suggestion. We checked the data and found that it was our translation problem, we apologize for this, and thank you for your careful pointing out. We have modified agouti. (Line28 and Line78).
- 18. line 64: amount
Response: Thank you very much for your suggestion. We have done the modification. (Line 69)
- line 55: the intake of
Response: Thank you very much for your suggestion. We have done the modification. (Line 55)
- line 67: molecular mechanism
Response: Thank you very much for your suggestion. We have done the modification.
- line 79: remove the
Response: Thank you very much for your suggestion. We have done the modification in line 78
- 22. line 252: why were these genes chosen in the list.
Response: Thank you very much for your suggestion. Our transcriptome data has only one set of data, and no biological replicates were performed to verify the accuracy of the data. The genes with |log2FC|≥1.5 and P<0.05 were selected from the gene data obtained in the transcriptome sequencing report, and genes related to fat metabolism were selected as far as possible. This is to make up for the lack of repeated experiments in the transcriptome, and to further prove that this set of transcriptome data is in high consistency with the actual regulation of cellular gene expression. (Line 179)
- line 270: all results are reversed? Why was this done like that?
Response: Thank you very much for your suggestion. We have reprocessed the data and regenerated transcriptome-related maps using KO-ASIP bMECs as the experimental group and WT bMECs as the control group.
- line 310: table says HacD4 is NS but here it says it is significant.
Response: Thank you very much for your suggestion. It has been mentioned before that our transcriptome sequencing has only performed one set of measurements, so the FDR value is high. Some genes are selected to verify the high FDR condition, and HACD4 is one of them. Our transcriptome sequencing results Validation is based on |log2FC|≥1.5 and P<0.05 as differential expression conditions. HACD4 meets this condition, but because the FDR of HACD4 does not meet the confidence interval of less than 0.05, it is not within the significant range of transcriptomics determined in the experiment , so it is normal that the final quantitative results are different from those screened in the transcriptome, but the expression trend of HACD4 is consistent.
- line 440: alteration should read alter
Response: Thank you very much for your suggestion. We have done the modification in line 493
- line 442: differentially expressed genes
Response: Thank you very much for your suggestion. We have done the modification in line 495

Reviewer 2 Report
The work entitled "The knockout of ASIP gene altered the lipid composition in bovine mammary epithelial cells via the expression of genes in lipid metabolism pathway" describe an interesting attempt to assess potential transcriptional biomarkers related to fatty acids composition in bovine milk. The present work represent therefore a good in vitro starting point to explore fatty acids metabolism and future selection of desiderable genetic traits in dairy cows.
Despite the work deserve pubblication, some revision are needed, considering both minor text check (for example "vitro model" instead of "vitro mode" at line 19) and better description of some technical aspects.
In general, when transcriptomics analysis are involved in a study, a clear description of biological and technical replicates of performed experiments are mandatory, in order to better justify a statistical significance (if any). I suggest the author to add a scheme and/or a table clearly reporting the number of biological (e.g cell lines) replicates and analysis (e.g PCR duplicate/triplicate) replicates.
Indeed, in the manuscript the authors claimed that for mean and standard error of the mean (SEM) calculations measurese were taken in triplicate (line 207), but when they described qPCR analysis they stated that all reactions were performed in duplicates (line 196).
Moreover, for a reliable evaluation of fold changes described in figure 4, I suggest to report the 95% Confidence Interval of reported fold changes, instead of SEM.
At line 192 authors report that 1 microgram of cDNA was loaded into PCR reactions. Please specify how the authors quantify cDNA, or if they referred to RNA amounts loaded into RT reactions.
Another critical aspects of gene expression studies is the choice of reference genes for normalization of gene expression data. The authors report a generic normalization of their data against Beta-actin (line 197). Why other reference genes and their relative evaluation by dedicated tools (Normfinder, Genorm, etc..) have not been considered by the authors? Please justify your choices.
In the statistical analysis section (line 206-207) please specify which post-hoc test was performed in their comparison study. A descriptive statistics should be also provide, in order to justify the choice of T-test instead of ANOVA and/or other non parametric test.
Author Response
Dear Reviewers
On behalf of all the authors of the submitted manuscript “The knockout of ASIP gene altered the lipid composition in bovine mammary epithelial cells via the expression of genes in lipid metabolism pathway”, we thank you so much for your kind reviews and meaningful suggestions. We have studied comments carefully and have made corrections which should meet with approvals we hope. The main revision in the paper (highlight changes in red) and the responses to the reviewers’ comments are as following:
Responds to Reviewer #2:
- Despite the work deserve pubblication, some revision are needed, considering both minor text check (for example "vitro model" instead of "vitro mode" at line 19) and better description of some technical aspects.
Response: Thank you very much for your suggestion. We have done the modification, see line 20
- In general, when transcriptomics analysis are involved in a study, a clear description of biological and technical replicates of performed experiments are mandatory, in order to better justify a statistical significance (if any). I suggest the author to add a scheme and/or a table clearly reporting the number of biological (e.g cell lines) replicates and analysis (e.g PCR duplicate/triplicate) replicates.
Response: Thank you very much for your suggestion. When we measured the transcriptome, we only did one group, and did not perform the corresponding biological replicates. To verify the accuracy of the transcriptome data of this group, we selected the transcriptome sequencing data that conformed to |log2FC|≥1.5 and P< 0.05 genes, and genes related to fat metabolism were selected as much as possible, and some non-fat related genes were randomly selected. This is to make up for the lack of repeated experiments in the transcriptome, and to further prove that this set of transcriptome data is in high consistency with the actual regulation of cellular gene expression.
- 3. Indeed, in the manuscript the authors claimed that for mean and standard error of the mean (SEM) calculations measurese were taken in triplicate (line 207), but when they described qPCR analysis they stated that all reactions were performed in duplicates (line 196).
Response: Thank you very much for your suggestion. We've made modifications to the unity, in fact, we've run out the transcriptomics experiment, the other experiments have been done with 3 biological replicates and 3 control replicates. (line197)
- Moreover, for a reliable evaluation of fold changes described in figure 4, I suggest to report the 95% Confidence Interval of reported fold changes, instead of SEM.
Response: Thank you very much for your suggestion. We have done the modification in figure 4 and lines 348-349.
- 5. At line 192 authors report that 1 microgram of cDNA was loaded into PCR reactions. Please specify how the authors quantify cDNA, or if they referred to RNA amounts loaded into RT reactions.
Response: Thank you very much for your suggestion. The amount of RNA loaded into the RT reaction is referred to here, which is our oversight. There is an error in the description. Thank you for pointing out our error. We have revised it at line193 in the article.
- Another critical aspects of gene expression studies is the choice of reference genes for normalization of gene expression data. The authors report a generic normalization of their data against Beta-actin (line 197). Why other reference genes and their relative evaluation by dedicated tools (Normfinder, Genorm, etc..) have not been considered by the authors? Please justify your choices.
Response: Thanks for your suggestion. The classic method is suitable too. There are many internal reference genes, such as β-actin, GAPDH, Tub, 18s rRNA, TBP et al. β-actin was used to be the internal reference gene in this experiment, because, our previous study has shown that β-actin can be stably expressed in bovine mammary epithelial cells. Thus, this experiment referred to the previous research and chose β-actin as the internal reference gene [21].
- In the statistical analysis section (line 206-207) please specify which post-hoc test was performed in their comparison study. A descriptive statistics should be also provide, in order to justify the choice of T-test instead of ANOVA and/or other non parametric test
Response: Thanks for your suggestion. The quantitative analysis of genes is assumed to obey the normal distribution. We carried out 3 biological replicates in the experiment, and in the analysis, the data of the 3 biological replicates of the experimental group and the 3 replicates of the control group were compared in pairs, so we need to compare two groups. the t-test prefered to used the two groups that why we selected t-test to showed the significance difference between the two groups. which we have modified and annotated in lines 208-211.
